# Barriers and facilitators to dementia care in long-term care facilities: protocol for a qualitative systematic review and meta-synthesis

Xi Zhang [ORCID],[1] Chengguo Guan,[2] Jinjie He,[3] Jing Wang[1]

[1]Faculty of Nursing, Health Science Center, Xi'an Jiaotong University, Xi'an, Shaanxi, China
[2]School of Nursing, The Open University of Shaanxi, Xi'an, Shaanxi, China
[3]School of Economics and Management, Xi'an University, Xi'an, Shaanxi, China

**Correspondence to**
Dr Jing Wang;
novowj@xjtu.edu.cn

## ABSTRACT

**Introduction** Long-term care needs for people with dementia are predicted to increase due to increased life expectancy and dementia diagnoses. Most published meta-syntheses of dementia care focus on hospitals or home settings. When focusing on long-term care facilities, most reviews about dementia care only focus on a single outcome, such as feeding, behavioural symptoms management, palliative care and others, which is limited. The present study aims to synthesise qualitative data and examine barriers and facilitators to caring for people with dementia in long-term care facilities.

**Methods and analysis** This is the protocol for our systematic review and meta-synthesis, which describes the design of this study, and we plan to complete the study from October 2023 to November 2024. The systematic review and meta-synthesis will follow the Joanna Briggs Institute (JBI) guidance for systematic reviews of qualitative evidence. Nine databases (five English and four Chinese) were searched, including Embase, Web of Science, Medline, CINAHL, PsycINFO and Wan Fang Data, China National Knowledge Infrastructure, VIP and Chinese Biomedical Medicine, from inception to August 2023. Qualitative and mixed-approach research about barriers and facilitators to caring for people with dementia in long-term care facilities, which are reported in English or Chinese, will be included. Covidence software will help with study selection, assessment and data extraction. The JBI Critical Appraisal Checklist for Qualitative Research (2020) will be used for included studies' quality assessment. Data extraction will be based on the JBI Qualitative Assessment and Review Instrument Data Extraction Tool for Qualitative Research. The JBI aggregation approach will be used to synthesise data. We will use the JBI ConQual tool to assess the credibility and dependability of each synthesised finding to establish confidence in the synthesised findings. All review steps will be managed by two reviewers independently, and disparities will be discussed. If consensus cannot reach a resolution, a third reviewer will be consulted.

**Ethics and dissemination** The present study is a secondary analysis of published qualitative data. So ethical approval is not required. The findings may be disseminated through peer-reviewed publications, conference papers or elsewhere.

**PROSPERO registration number** The protocol was registered with the International Prospective Register of

## STRENGTHS AND LIMITATIONS OF THIS STUDY

⇒ This study will be guided by the Joanna Briggs Institute methodology for qualitative systematic reviews to increase the accuracy and reliability of results.
⇒ The consensus-based approach to analysis will ensure shared responsibility for interpretative decisions and the rigour and trustworthiness of the findings of this study.
⇒ A limitation is the exclusion of studies not written in English or Chinese, and of grey literature.

Systematic Reviews (PROSPERO) in May 2022, and the registration number is CRD42022326178.

## INTRODUCTION

Dementia is a degenerative syndrome affecting individuals' cognitive functions. It impacts memory, behaviour, thinking and social abilities as the disease advances. Those affected become increasingly dependent on care from others for all activities of daily living.[1] Previous studies show that age is a critical factor in dementia, and there is no cure for dementia currently.[2] The demographic shift to an ageing population has rapidly increased the number of people with dementia. According to the most recent WHO figures, 55.2 million people worldwide were living with dementia in 2019 and the number is expected to rise to 139 million in 2050.[3] Caring for people with dementia requires round-the-clock care, and caregivers must manage behavioural and psychological symptoms (such as aggression, wandering and depression), placing an overwhelming burden on the caregivers' well-being, especially in the severe stage of dementia.[4][5] Long-term care facilities are the alternative place when people with dementia cannot get sufficient support at home.[6][7] Long-term care facilities may have different names

in different countries, such as nursing homes, skilled nursing facilities, assisted living facilities and residential facilities, providing services to people who cannot live independently.[8] Approximately, over 30% of long-term care facility residents have a form of dementia in higher-income economies.[3] In lower-income and middle-income countries, informal caregivers, mostly family members, provide a significant proportion of dementia care nowadays.[3] However, long-term care needs for people with dementia are predicted to increase due to increased life expectancy and dementia diagnoses.

The WHO Framework for Countries to achieve an integrated continuum of long-term care recognises critical aspects for an integrated continuum service provision in long-term care facilities and guides countries in evaluating and implementing equitable and sustainable care in long-term care facilities.[9] The goal of long-term care is to ensure that an individual who has significant declines in physical or mental capacity can maintain the best possible quality of life, with the greatest possible degree of independence, autonomy, participation, personal fulfilment and human dignity.[9] Professionals need to take a holistic approach to caring for people with dementia. Depending on the situation, dementia care should include basic care such as eating and dressing, pain management, behavioural and psychological symptoms management and palliative care to promote well-being and quality of life.[10] However,there is still a gap between the theory and practice of dementia care in long-term care facilities. Studies suggest that the quality of life of people with demnetia in facilities still needs to be improved,[11] and that feeding difficulties,[12] pain management[13] and other disease-related care needs require further attention. Besides, behavioural and psychological symptoms are common in people with dementia, and how to manage these symptoms is still an endeavour.[14] Professionals, as key service providers in long-term care facilities, face many difficulties in providing care for people with dementia.[15–18] Some qualitative studies suggest that professionals find it hard to provide personalised care and interact with people in long-term care facilities because of insufficient time and a lack of knowledge about dementia.[15 17] It has also been noted that when a professional's attitude towards dementia is positive and hopeful, the more it contributes to the caring of people with dementia.[19] In addition to exploring the experiences of professionals in holistic dementia care, there are also studies that focus on a single outcome. Some studies suggest that management of behavioural and psychological symptoms may be limited by lack of clinical guidance and resources,[20] and feeding care is also associated with clinical resources, in addition to the environment.[21]

In recent years, qualitative studies have proliferated in nursing research because they effectively complement quantitative research results.[22] Qualitative studies explore detailed explanations and meanings shown by participants, providing an alternative perspective to a more positivist approach.[23] Meta-synthesis of qualitative studies can yield broader results because they synthesise a more comprehensive range of participants and descriptions. It integrates the body of knowledge about a topic and, as a result of this process, can produce new interpretations of the phenomenon that contribute to understanding the topic.[24] To date, a limited number of qualitative studies have explored the experience of dementia care from the perspective of professionals. There have been some meta-analyses based on these articles, but most published meta-syntheses of dementia care focus on hospital[25–27] or home settings.[28–30] When focusing on long-term care facilities, most reviews about dementia care only focus on a single outcome, such as feeding,[31 32] behavioural symptoms management,[33 34] palliative care[35] and others, which is limited. The dementia care is holistic but considering only one aspect of care is not comprehensive. Integrating these known factors to identify barriers and facilitators to dementia care in long-term care facilities is essential.

To the best of our knowledge, there are currently no studies that have systematically evaluated qualitative research on holistic dementia care in long-term care facilities. Therefore, the present study aims to systematically review and synthesise the best available qualitative evidence on barriers and facilitators to caring for people with dementia in long-term care facilities from the perspective of the professionals, providing more comprehensive evidence for better service development in dementia care in long-term care facilities.

## METHOD AND ANALYSIS
### Design
The protocol was registered with the International Prospective Register of Systematic Reviews (PROSPERO) in May 2022, and the registration number is CRD42022326178. This protocol was designed following the Preferred Reporting Items for Systematic Review and Meta-Analysis Protocols (PRISMA-P) 2015 statement[36 37] and the Joanna Briggs Institute (JBI) guidance for systematic reviews of qualitative evidence.[38] The study will run from October 2023 to November 2024.

### Eligibility criteria
#### Type of participants
Studies focusing on health professionals (aged≥18) caring for people with dementia in long-term care facilities, including doctors, nurses and other health professionals. Studies focusing on multiple disease types (not only care for people with dementia) will also be excluded.

#### Phenomena of interest
Studies focusing on barriers and facilitators to caring for people with dementia in long-term care facilities, including experiences in basic care such as eating and dressing, pain management, behavioural and psychological symptoms management, palliative care and other-related care.

## Context

Studies focusing on long-term care facilities,[8] such as nursing homes, skilled nursing facilities, assisted living facilities, nursing institutes and residential facilities, providing services to people who cannot live independently. Studies conducted in adult daycare facilities, respite care and other transitional care facilities will also be included.

## Types of studies

Qualitative and mixed methods studies with an analytical part of the qualitative content published in English or Chinese will be included. Only the qualitative component of the mixed methods studies will be considered. Quantitative studies, pilot studies, reviews, case reports, case series, reanalysis of data, conference abstracts and expert opinions will be excluded.

## Information sources

Nine databases will be searched, including five English databases and four Chinese databases with no time limit applied: The five English databases are Embase, Web of Science, Medline, CINAHL, PsycINFO and the four Chinese databases are Wan Fang Data, China National Knowledge Infrastructure (CNKI), VIP and Chinese Biomedical Medicine. References and citations of the included studies will also be searched to identify if there are any other relevant sources.

## Search strategy

Keywords were first formed based on the eligibility criteria, and synonyms of each keyword were then analysed to create a logic grid to capture relevant studies. Boolean operators will be used to combine all the keywords. Pilot searches will be conducted to confirm the sensitivity and specificity of the strategy in all databases before the formal search is conducted. The final strategy will be consulted by an expert librarian and reviewed by all team members. The index terms and the search strategy were developed in Web of Science among five English databases and CNKI among four Chinese databases. The Web of Science English search strategy and CNKI Chinese search strategy will be then translated and used for additional databases. We will include studies retrievable from the establishment of the database to August 2023. Search strategies are available in online supplemental file 1. Citation tracking will also be conducted, with the reference lists of included studies being screened to identify studies that were missed in the search process.

## Study selection

All search records will be imported to Endnote X9 (V.9.3.3) for duplicates removal and then to Covidence software (https://app.covidence.org/reviews/active) for study selection. Covidence is a systematic review production tool developed by researchers specialised in the systematic review process to standardise the process of conducting systematic reviews. According to the proposed eligibility criteria, title and abstract screening will be conducted by two reviewers (CG and XZ) independently on all records. Then, the full-text review of included studies will be screened in the same way, and the reason for exclusion will be selected and recorded in Covidence. Before each formal process, a pilot test will be conducted initially with 5%–10% of studies screened by all reviewers independently to facilitate understanding and application of eligibility criteria and the tool. Disparities will be discussed during the screening process via group discussion. If a resolution cannot be reached by consensus, the third and fourth reviewers (JH and JW) will be consulted. Regular group meetings with all the reviewers will be held regularly to discuss and resolve any disagreements. The results of the screening will be recorded in a PRISMA flow chart.

## Risk-of-bias assessment

The JBI Critical Appraisal Checklist for Qualitative Research (2020) (online supplemental file 2) will be used for assessing the quality of included studies, which includes 10 items that assess the studies' philosophical foundation, methodology, method of data collection, method of data analysis, results interpretation, the logic of studies and research ethics. The four responses 'yes,' 'no,' 'unclear' and 'not applicable' will be used to evaluate each item. The final quality rating of the study will be rated as high, medium or low based on the assessment of individual items and the amount of weight given to the degree to which each item impacted the findings of the study.[38 39] Only studies having at least a medium ranking will be included. Two reviewers (CG and XZ) will appraise all included studies' methodological quality independently and disparities will be discussed. The third and fourth reviewers (JH and JW) will be consulted when a resolution cannot be reached by consensus.

## Data extraction

Data extraction will be conducted based on the JBI Qualitative Assessment and Review Instrument Data Extraction Tool for Qualitative Research (online supplemental file 3) by two reviewers (CG and XZ) independently.[38] According to the tool, the characteristics of included studies and findings with illustrations will be extracted. Information will be extracted using Covidence, including (1) author, publication, year, country, geographical location and study setting; (2) participants' characteristics and sample size; (3) research design, methodology and methods ; (4) phenomena of interest and main findings. Before the commencement of the formal data extraction, data from 5% to 10% of the included studies will be extracted using the data extraction tool to pilot test the tool and minimise the risk of error. According to the JBI review manual, all the findings extracted from the selected studies will be evaluated and divided into three levels based on the degree of support between the finding and its illustration: unequivocal, credible and unsupported.[38] Only the findings rated as unequivocal or credible will be included for data synthesis. Any disagreements between the two

reviewers will be discussed and resolved by regular team meetings with the four reviewers.

## Data synthesis

The JBI meta aggregation approach with a three-step process will be used for data synthesis after data extraction.[40] In the three-step data synthesis process, the findings will first be extracted from the original studies, then the included findings will be grouped into categories based on similarity of meaning and similar categories will ultimately be further synthesised into different synthesised findings based on their relationships and relevance to the research objective of this review. This approach focuses on the essence of qualitative research and emphasises the value and role of qualitative research in an evidence-based healthcare service system, collecting all included studies' findings and summarising them according to their meanings to make them more targeted, convincing and generalising. On the premise of understanding the philosophical foundation and methodology of various qualitative studies, reviewers repeatedly read, understand, analyse and interpret the meaning of each included findings, combine similar findings, form a new category and finally summarise the category as synthesised findings. In the present review, the findings of each included study will be read carefully to understand completely before data synthesis. Regular meetings will be held with the four reviewers to reach an agreement on the final synthesised findings, and any conflicting opinions between the reviewers will be settled by thorough discussions.

## Assessing the confidence of findings

We will use the JBI ConQual tool to assess the confidence of each synthesised finding.[41] Dependability and credibility are two elements that influence the confidence of qualitative synthesised findings. Dependability can be established by examining the quality of the original studies included through a set of critical appraisal questions. Credibility evaluates the goodness of fit between the author's interpretation and the original data. First, the confidence of synthesised findings will be initially assumed to be graded as high, and then downgraded based on the results of dependability and credibility assessments, and finally each synthesised finding will be rated high, moderate, low or very low.[41]

## ETHICS AND DISSEMINATION

The present study is a secondary analysis of published qualitative data; there is no concern for privacy. So ethical approval is not required for the present study. However, the research ethics of each included original study will be assessed rigorously. The findings may be disseminated through peer-reviewed publications, conference papers or elsewhere.

**Acknowledgements** We express our gratitude for the support of the National Social Science Fund of China.

**Contributors** JW and CG conceived the topic of this study. All team members (JW, CG, JH and XZ) participated in the discussion of the study design and made significant comments. JW and CG constructed the search strategy. CG, JH, XZ and JW will conduct data selection, assessment, extraction and synthesis. XZ drafted the protocol manuscript, and then the manuscript was reviewed by JW. All authors approved the submitted protocol and are accountable for its content.

**Funding** This work was supported by the National Social Science Fund of China (grant number 20XRK004).

**Competing interests** None declared.

**Patient and public involvement** Patients and/or the public were not involved in the design, or conduct, or reporting, or dissemination plans of this research.

**Patient consent for publication** Not applicable.

**Provenance and peer review** Not commissioned; externally peer reviewed.

**ORCID iD**
Xi Zhang http://orcid.org/0009-0000-4217-3688

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
