## [Reviewer comments · BMJ Open]

ARTICLE DETAILS

TITLE (PROVISIONAL)	Barriers and facilitators to dementia care in long-term care facilities: Protocol for a qualitative systematic review and meta-synthesis
AUTHORS	Zhang, Xi; Guan, Chengguo; He, Jinjie; Wang, Jing

VERSION 1 – REVIEW

REVIEWER	Heinrich, Steffen Eastern Switzerland University of Applied Sciences
REVIEW RETURNED	14-Jun-2023

GENERAL COMMENTS	The topic that the review addresses is highly relevant. Basically, the methodology is solidly thought out. However, there are some questions and limitations that need to be addressed or at least explained: Eligibility criteria: A table regarding inclusion criteria would significantly increase clarity. The assessment of facilitating and inhibiting factors for the implementation of dementia care also includes assessments of relatives and people with dementia themselves. Often a bias arises when only the view of professionals is included. Why is the exclusion of PwD and relatives planned? (Line 138) Search strategy: Is there a reason why citation tracking is not planned? This could be done via Scopus, for example. Search String: There is no subdivision into Title/Abstract and MESH search terms in the search string. This should be done for a comprehensive search with the highest possible search quality. Furthermore, there are multiple duplications in the search terms (e.g. nursing home). I recommend reducing the redundancies in this respect and checking whether so many keywords per search term are all target-oriented. Risk of bias assessment: As far as I know, one cannot simply determine the study quality (high, medium, low) in JBI by taking the number of fulfilled items as a purely quantitative basis. This is not permissible, but always an individual decision process, since individual items can have more or less weight depending on the study. Data extraction: In addition to piloting the JBI-QARI (Lines 207 - 209), I recommend that 5%-10% of found titles and abstracts as well as full texts will be
--

	screened independently by all reviewers. Such calibration will help to understand and apply the eligibility criteria.
--	---

REVIEWER	Bray, Jennifer University of Worcester
REVIEW RETURNED	07-Aug-2023

GENERAL COMMENTS	Lines 35-36 – rather than ‘caring for dementia’ I would suggest something like ‘caring for people with dementia’ or ‘providing dementia care’ Same applies to line 43. I appreciate that the article title says it’s a protocol, but it’s probably worth making it clear in the abstract what the timeframe is (June 2022 – March 2024) to avoid potential confusion, as I initially found it odd that you were saying what you ‘will be’ doing. Maybe also rewording the second half of title to 'Protocol for a systematic review and meta-synthesis' might draw more attention to it. Line 76 – I don’t think you need the phrase ‘are about to’ I know that you give your reasons for why your systematic review is needed, but the fact that you list a whole series of barriers and facilitators in lines 101-105 does make me wonder slightly if your review is really going to add much. I look forward to seeing what additional evidence and information you identify. Eligibility criteria – I find it odd that you talk about ‘studies reporting on’ and ‘studies covering’ but then you say ‘studies focus on’. Surely for consistency you should say ‘studies focusing on’? Line 143 – ‘long-term’ not ‘Long-term’ Line 188 – you use JBI but you haven’t said what that stands for. You did in the abstract, but not in the main article. Lines 235-237 – you talk about assessing ‘dependability and credibility’ but then explain what ‘dependability’ and ‘reliability’ are. You also further talk about ‘stability and credibility assessments’. What are you actually assessing? A bit of consistency in your terminology would be useful.
---

REVIEWER	Wiseman, Jodie Univ Queensland, School of Health and Rehabilitation Sciences
REVIEW RETURNED	06-Sep-2023

GENERAL COMMENTS	An interesting topic with the potential to address an important gap in the literature. A few points to consider:  1. Title: The JBI manual for evidence synthesis advises including the term "qualitative systematic review protocol" in the title to emphasise it will be reviewing qualitative research 2. Introduction The third paragraph covers multiple topics - may benefit from re-structuring to improve clarity May benefit from a statement re: sources searched to determine if a systematic review in this area exists as per the JBI manual It is not clear what area/s of dementia care is being addressed - it states an integration of factors is important but this does not clearly link with the following statement re: barriers and facilitators 3. Methods Clear and reproducible methods covering important aspects of JBI and PRISMA P.
---

	As per above comment the Phenomena of interest could be clarified May benefit from an updated search period to determine if any new articles have been published since May 2022.
--	--

VERSION 1 – AUTHOR RESPONSE

Response to Reviewer #1' comments

Dear Dr. Steffen Heinrich,

Thank you very much for your time reviewing the manuscript and encouraging comments on the merits: "The topic that the review addresses is highly relevant. Basically, the methodology is solidly thought out." We appreciate your clear and detailed feedback and hope the explanation has fully addressed your concerns. In this letter, we discussed each of your comments individually along with our corresponding responses. To facilitate this discussion, our responses to your comments are highlighted in blue, and modifications or additions are highlighted in red. You can see a clear copy (without highlighted changes) of the revised manuscript in "Main document" and an edited version of the revised manuscript in "Main document – marked copy".

Comment 1:

Eligibility criteria:

A table regarding inclusion criteria would significantly increase clarity.

The assessment of facilitating and inhibiting factors for the implementation of dementia care also includes assessments of relatives and people with dementia themselves. Often a bias arises when only the view of professionals is included. Why is the exclusion of PwD and relatives planned? (Line 138)

Response1: Thank you very much for your invaluable comments. To address your comments, we have developed a table presenting inclusion and exclusion criteria. However, upon further review, we found a lot of duplication between the table and the text. Furthermore, in accordance with the requirements of the journal, we eventually removed the table. However, per your suggestion, we have refined the Eligible Criteria section to clarify it.

Regarding the comment "Why is the exclusion of PwD and relatives planned?" here are our responses: Considering that professionals are the primary providers of care services in long-term care facilities, we have focused our perspective on them primarily, and subsequent research may further explore the perceptions of PwD and relatives. In addition, based on your suggestions, we have also reviewed the "Induction" section and added some relevant content to ensure logical coherence. (Line 112-122) "...Professionals, as key service providers in long-term care facilities, face many difficulties in providing care for people with dementia.[15-18] Some qualitative studies suggest that professionals

find it hard to provide personalized care to patients and interact with them in long-term care facilities because of insufficient time and a lack of knowledge about dementia.[15, 17] It has also been noted that when a professional's attitude towards dementia is positive and hopeful, the more it contributes to the caring of people with dementia.[19] In addition to exploring the experiences of professionals in holistic dementia care, there are also studies that focus on a single outcome. Some studies suggest that management of behavioral and psychological symptoms may be limited by lack of clinical guidance and resources,[20] and feeding care is also associated with clinical resources, in addition to the environment.[21]"

Comment 2:

Search strategy:

Is there a reason why citation tracking is not planned? This could be done via Scopus, for example.

Response 2: Thank you for your comments. In our original manuscript, citation tracking would be conducted in the studies that were included after full-text review. According to your comments, we have verified our methodology and plan to conduct citation tracking after the steps of the literature search to ensure comprehensiveness. (Line 193-195) "...Besides, citation tracing will be conducted, the reference lists of included studies will be screened to identify studies that were missed in the search process."

Comment 3:

Search String:

There is no subdivision into Title/Abstract and MESH search terms in the search string. This should be done for a comprehensive search with the highest possible search quality. Furthermore, there are multiple duplications in the search terms (e.g. nursing home). I recommend reducing the redundancies in this respect and checking whether so many keywords per search term are all target-oriented.

Response 3: Thank you for your comments. Except for databases that do not provide MeSH search services (e.g., WoS), we performed MeSH searches in all other databases, and the search queries are bolded in Appendix 1. In addition, except for the MEDLINE database (which has fewer relevant articles with Title/Abstract search terms), all other databases were searched using Title/Abstract search terms. The TI/AB search field identifications were bolded in the search strategy in Appendix 1.

Moreover, many thanks for raising the issue of duplications in the search terms. We have double-checked all the search terms and removed duplicates.

Comment 4:

Risk of bias assessment:

As far as I know, one cannot simply determine the study quality (high, medium, low) in JBI by taking

the number of fulfilled items as a purely quantitative basis. This is not permissible, but always an individual decision process, since individual items can have more or less weight depending on the study.

Response 4: Thank you for your comments. Per your suggestion, we have reviewed the JBI manuals and relevant literature again and have revised this section accordingly. (Line 218-220) "...The final quality rating of the study would be rated as high, medium, or low based on the assessment of individual items and the amount of weight given to the degree to which each item impacted the findings of the study.[38, 39]..."

Comment 5:

Data extraction:

In addition to piloting the JBI-QARI (Lines 207 - 209), I recommend that 5%-10% of found titles and abstracts as well as full texts will be screened independently by all reviewers. Such calibration will help to understand and apply the eligibility criteria.

Response 4: Thank you for your suggestion. We have revised our manuscript accordingly. (Line 204-207) "...Before the commencement of the formal data extraction, data of 5-10 percent included studies will be extracted using the data extraction tool to pilot test the tool and minimize the risk of error. ..." In addition, in the "Data extraction" section, we standardized the number of studies included in the pilot test, changing it from two in the original plan to five to ten percent. (Line 233-235) "...Before the commencement of the formal data extraction, data of 5-10 percent included studies will be extracted using the data extraction tool to pilot test the tool and minimize the risk of error. ..."

Reply to Reviewer #2

Dear Ms. Jennifer Bray,

Thank you very much for your time involved in reviewing the manuscript and your invaluable comments to help us improve our manuscript. We appreciate your clear and detailed feedback and hope the explanation has fully addressed your concerns. In this letter, we discussed each of your comments individually along with our corresponding responses. To facilitate this discussion, our responses to your comments are highlighted in blue, and modifications or additions are highlighted in red. You can see a clear copy (without highlighted changes) of the revised manuscript in "Main document" and an edited version of the revised manuscript in "Main document – marked copy".

Comment 1: Lines 35-36 – rather than 'caring for dementia' I would suggest something like 'caring for people with dementia' or 'providing dementia care'

Same applies to line 43.

Response 1: Thank you very much for pointing this out. We have checked the whole manuscript and

revised the original expression “caring for dementia” to “caring for people with dementia” (Line 35, 44). Besides, we verified the expression elsewhere.

Comment 2: I appreciate that the article title says it’s a protocol, but it’s probably worth making it clear in the abstract what the timeframe is (June 2022 – March 2024) to avoid potential confusion, as I initially found it odd that you were saying what you ‘will be’ doing. Maybe also rewording the second half of title to 'Protocol for a systematic review and meta-synthesis' might draw more attention to it.

Response 2: Thank you for raising this important issue. We have added the timeframe in the abstract to make it clearer. (Line 36-38) “...This is the protocol for our systematic review and meta-synthesis, which describes the design of this study, and we plan to complete the study from October 2023 to November 2024...”. In addition to, we have revised the title. (Line 1-2) “Barriers and facilitators to dementia care in long-term care facilities: Protocol for a qualitative systematic review and meta-synthesis”.

Comment 3: Line 76 – I don’t think you need the phrase ‘are about to’

Response 3: Thank you for your comments. We have deleted “are about to” (Line 77) accordingly. We also verified the expression elsewhere.

Comment 4: I know that you give your reasons for why your systematic review is needed, but the fact that you list a whole series of barriers and facilitators in lines 101-105 does make me wonder slightly if your review is really going to add much. I look forward to seeing what additional evidence and information you identify.

Response 4: Thank you for your comments. We have revised the “Introduction” section and made some changes to make the logic of this study clearer. We added content about the current status of care for people with dementia in long-term care facilities and the limitations of existing research. The relevant contents are provided below as a screen dump for your quick reference. The modified and added contents have been marked in red. (Line 96-146)

96 The WHO Framework for Countries to achieve an integrated continuum of long-term
97 care recognizes critical aspects for an integrated continuum service provision in long-
98 term care facilities and guides countries in evaluating and implementing equitable and
99 sustainable care in long-term care facilities.[9] The goal of long-term care is to ensure
100 that an individual who has significant declines in physical or mental capacity can
101 maintain the best possible quality of life, with the greatest possible degree of
102 independence, autonomy, participation, personal fulfillment and human dignity.[9]
103 Professionals need to take a holistic approach to providing care to the people living
104 with dementia, depending on the situation, dementia care should include basic care such
105 as eating and dressing, pain management, behavioral and psychological symptoms
106 management, and palliative care to promote their well-being and quality of life.[10] But
107 now, knowledge-practice gaps about dementia care in long-term care facilities have
108 been still revealed by several studies: the quality of life still need to be improved;[11]
109 feeding difficulties,[12] pain management,[13] and other disease-related care needs
110 require further attention. Besides, the behavioral and psychological symptoms are
111 common in people with dementia, and how to manage these symptoms is still an
112 endeavor.[14] Professionals, as key service providers in long-term care facilities, face
113 many difficulties in providing care for people with dementia.[15-18] Some qualitative
114 studies suggest that professionals find it hard to provide personalized care to patients
115 and interact with them in long-term care facilities because of insufficient time and a _____

116 lack of knowledge about dementia.[15, 17] It has also been noted that when a
117 professional's attitude towards dementia is positive and hopeful, the more it contributes
118 to the caring of people with dementia.[19] In addition to exploring the experiences of
119 professionals in holistic dementia care, there are also studies that focus on a single
120 outcome. Some studies suggest that management of behavioral and psychological
121 symptoms may be limited by lack of clinical guidance and resources,[20] and feeding
122 care is also associated with clinical resources, in addition to the environment.[21]

123 In recent years, qualitative studies are proliferating in nursing research because they
124 effectively complement quantitative research results.[22] Qualitative studies explore
125 detailed explanations and meanings shown by participants, providing an alternative
126 perspective to a more positivist approach.[23] Meta-synthesis of qualitative studies can
127 yield broader results because they synthesize a more comprehensive range of
128 participants and descriptions. It integrates the body of knowledge about a topic and, as
129 a result of this process, can produce new interpretations of the phenomenon that
130 contribute to understanding the topic.[24] To date, a certain number of qualitative
131 studies have explored the experience of dementia care from the perspective of
132 professionals. There has been some meta-analysis based on these articles, but most

133 published meta-synthesis of dementia care focus on hospital[25-27] or home
134 settings.[28-30] Regarding long-term care facilities, most reviews about dementia care
135 only focus on a single outcome, such as feeding,[31, 32] behavioral symptoms
136 management,[33, 34] palliative care,[35] and others, which is limited. The dementia
137 care is holistic, and considering only one aspect of care is not comprehensive.
138 Integrating these known factors to identify barriers and facilitators to dementia care in
139 long-term care facilities is essential.

140 To the best of our knowledge, there are currently no studies that have systematically
141 evaluated qualitative research on holistic dementia care in long-term care facilities.

142 Therefore, the present study aims to systematically review and synthesise the best
143 available qualitative evidence on barriers and facilitators to caring for people with
144 dementia in long-term care facilities from the perspective of the professionals,
145 providing more comprehensive evidence for better service development in dementia
146 care in long-term care facilities.

Comment 5: Eligibility criteria – I find it odd that you talk about ‘studies reporting on’ and ‘studies covering’ but then you say ‘studies focus on’. Surely for consistency you should say ‘studies focusing on’?

Response 5: Thank you for your comments. We have changed all inconsistent expressions to “Studies focus on”. (Line 157, 159, 161, 165) “...**Studies focusing on**...”. Furthermore, we have checked the whole manuscript to avoid similar problems.

Comment 6: Line 143 – ‘long-term’ not ‘Long-term’

Response 6: Thank you for pointing this out. We have corrected the error accordingly. (Line 162) “...**long-term care facilities**...”.

Comment 7: Line 188 – you use JBI but you haven’t said what that stands for. You did in the abstract, but not in the main article.

Response 7: Thank you for raising this issue. We have added the full name of JBI when it first appeared in the main article. (Line 154) “...**Joanna Briggs Institute (JBI)**...”.

Comment 8: Lines 235-237 – you talk about assessing ‘dependability and credibility’ but then explain what ‘dependability’ and ‘reliability’ are. You also further talk about ‘stability and credibility assessments’. What are you actually assessing? A bit of consistency in your terminology would be useful.

Response 8: Thank you for your invaluable comments. We have revised the “Assessing the confidence of findings” section to address your comments. The terminology was used consistently. The relevant contents are provided below as a screen dump for your quick reference. The modified and added contents have been marked in red. (Line 261-270) “...**We will use the JBI ConQual tool to assess the confidence of each synthesised finding.[41] Dependability and credibility are two elements to influence the confidence of qualitative synthesised findings. Dependability can be established by examining the quality of the original studies included through a set of critical appraisal questions. Credibility evaluates the goodness of fit between the author's interpretation and the original data. First, the confidence of synthesised findings will be initially assumed to be graded as high, and then downgraded based on the results of dependability and credibility assessments, and finally each synthesised finding will be rated high, moderate, low, or very low.[41]**”

Response to Reviewer #3’s comments

Dear Ms. Jodie Wiseman,

Thank you very much for your time involved in reviewing the manuscript and your encouraging comments on the merits: "An interesting topic with the potential to address an important gap in the literature." We appreciate your clear and detailed feedback and hope the explanation has fully addressed your concerns. In this letter, we discussed each of your comments individually along with our corresponding responses. To facilitate this discussion, our responses to your comments are highlighted in blue, and modifications or additions are highlighted in red. You can see a clear copy (without highlighted changes) of the revised manuscript in "Main document" and an edited version of the revised manuscript in "Main document – marked copy".

Comment 1:

Title: The JBI manual for evidence synthesis advises including the term "qualitative systematic review protocol" in the title to emphasise it will be reviewing qualitative research

Response 1: Thank you for pointing this out. We have revised the title accordingly. (Line 1-2) "Barriers and facilitators to dementia care in long-term care facilities: Protocol for a qualitative systematic review and meta-synthesis".

Comment 2:

Introduction

The third paragraph covers multiple topics - may benefit from re-structuring to improve clarity

May benefit from a statement re: sources searched to determine if a systematic review in this area exists as per the JBI manual

It is not clear what area/s of dementia care is being addressed - it states an integration of factors is important but this does not clearly link with the following statement re: barriers and facilitators

Response 2: Thank you so much for your comments. Per your suggestion, we have reviewed the original manuscript's third paragraph and re-structured it to make it more transparent. The relevant contents are provided below as a screen dump for your quick reference. The modified and added contents have been marked in red. (Line 96-122)

96 The WHO Framework for Countries to achieve an integrated continuum of long-term
97 care recognizes critical aspects for an integrated continuum service provision in long-
98 term care facilities and guides countries in evaluating and implementing equitable and
99 sustainable care in long-term care facilities.[9] The goal of long-term care is to ensure
100 that an individual who has significant declines in physical or mental capacity can
101 maintain the best possible quality of life, with the greatest possible degree of
102 independence, autonomy, participation, personal fulfillment and human dignity.[9]
103 Professionals need to take a holistic approach to providing care to the people living
104 with dementia, depending on the situation, dementia care should include basic care such
105 as eating and dressing, pain management, behavioral and psychological symptoms
106 management, and palliative care to promote their well-being and quality of life.[10] But
107 now, knowledge-practice gaps about dementia care in long-term care facilities have
108 been still revealed by several studies: the quality of life still need to be improved;[11]
109 feeding difficulties,[12] pain management,[13] and other disease-related care needs
110 require further attention. Besides, the behavioral and psychological symptoms are
111 common in people with dementia, and how to manage these symptoms is still an
112 endeavor.[14] Professionals, as key service providers in long-term care facilities, face
113 many difficulties in providing care for people with dementia.[15-18] Some qualitative
114 studies suggest that professionals find it hard to provide personalized care to patients
115 and interact with them in long-term care facilities because of insufficient time and a
116 lack of knowledge about dementia.[15, 17] It has also been noted that when a
117 professional's attitude towards dementia is positive and hopeful, the more it contributes
118 to the caring of people with dementia.[19] In addition to exploring the experiences of
119 professionals in holistic dementia care, there are also studies that focus on a single
120 outcome. Some studies suggest that management of behavioral and psychological
121 symptoms may be limited by lack of clinical guidance and resources,[20] and feeding
122 care is also associated with clinical resources, in addition to the environment.[21]

In addition to, we have also added the content (Line 140-141): "...To the best of our knowledge, there are currently no studies that have systematically evaluated qualitative research on holistic dementia care in long-term care facilities. ...".

Regarding the comment “It is not clear what area/s of dementia care is being addressed”, we have revised and added some content to improve the clarity of this section. The relevant contents are provided below as a screen dump for your quick reference. The modified and added contents have been marked in red. (Line 130-139)

130 contribute to understanding the topic.[24] To date, a certain number of qualitative
131 studies have explored the experience of dementia care from the perspective of
132 professionals. There has been some meta-analysis based on these articles, but most
133 published meta-synthesis of dementia care focus on hospital[25-27] or home
134 settings.[28-30] Regarding long-term care facilities, most reviews about dementia care
135 only focus on a single outcome, such as feeding,[31, 32] behavioral symptoms
136 management,[33, 34] palliative care,[35] and others, which is limited. The dementia
137 care is holistic, and considering only one aspect of care is not comprehensive.
138 Integrating these known factors to identify barriers and facilitators to dementia care in
139 long-term care facilities is essential.

Comment 3:

Methods

Clear and reproducible methods covering important aspects of JBI and PRISMA P.

As per above comment the Phenomena of interest could be clarified

May benefit from an updated search period to determine if any new articles have been published since May 2022.

Response 3: Thank you for your comments. Based on the changes made in response 2, we have clarified the phenomena of interest. We have revised the “Phenomena of interest” (Line 161-164)

“Phenomena of interest: Studies focusing on barriers and facilitators to caring for people with dementia in long-term care facilities, including experiences in basic care such as eating and dressing, pain management, behavioral and psychological symptoms management, palliative care and other related care.”

In addition, we have updated the search period. (Line 43,192) “...from inception to August 2023.”

VERSION 2 – REVIEW

REVIEWER	Heinrich, Steffen Eastern Switzerland University of Applied Sciences
REVIEW RETURNED	26-Sep-2023

GENERAL COMMENTS	I have no further acute revision requirements to this manuscript. However, I would still include the voices of some people with dementia on the topic. I find the rationale for non-inclusion to be of limited validity. With regard to potential resource limitations, however, it would be justifiable with reference to the need for further research on the topic, including this target group.
---

REVIEWER	Bray, Jennifer University of Worcester
REVIEW RETURNED	28-Sep-2023

GENERAL COMMENTS	Thank you for taking on board the review comments and revising your article. I can see that you have put a lot of effort into it. Unfortunately the changes have introduced a few issues that I think need to be addressed, such as long, confusing sentences or wording that does not make sense. I've raised these issues below. Abstract, lines 30-33 – the sentence 'Most published...is limited' is very long and confusing. I can't work out what it is saying. Which of the following is correct? Most published meta-synthesis of dementia care focus on hospital or home settings, or long-term care facilities. Most reviews about dementia care only focus on a single outcome... Or Most published meta-synthesis of dementia care focus on hospital or home settings. When focusing on long-term care facilities, most reviews about dementia care only focus on a single outcome... It's probably worth rewriting and splitting the sentence to make it clearer. Lines 103-106 – the sentence 'Professionals need...quality of life' is also very long and confusing. Do you mean: Professionals need to take a holistic approach to providing care to the people living with dementia, depending on the situation. Dementia care should include... Or Professionals need to take a holistic approach to providing care to the people living with dementia. Depending on the situation, dementia care should include... Is the phrase 'depending on the situation' actually needed? Line 106 – I don't 'their' is needed. You can probably just say 'to promote well-being' Lines 106-110 – I think this sentence might sound better if it is slightly reworded to something like: 'However, knowledge-practice gaps about dementia in long-term care facilities have been revealed by several studies, such as quality of life still needing to be improved,[11] while feeding difficulties,[12] pain management,[13] and other disease-related care needs require further attention.' Line 110 – I don't think 'the' is needed. You can probably just say 'Besides, behavioral' Lines 114-115 – it might be better to say 'to provide personalized care and interact with people in long-term' – it avoids referring to people at 'patients' and 'them' Line 130 – rather than 'a certain number of' I'd suggest saying 'a limited number of' or 'a few' Line 137 – 'holistic, but' might be better than 'holistic, and' Lines 193-195 – this sentence might sound better as 'Citation tracing will also be conducted, with the reference lists of included studies being screened to identify studies that were missed in the search process.' Line 205 – 'a pilot test' might be better than 'the pilot test'
---

	Line 218 – ‘will be rated’ rather than ‘would be rated’ Lines 233-234 – ‘data from 5-10 percent of the included’ rather than ‘data of 5-10 percent included’
--	---

VERSION 2 – AUTHOR RESPONSE

Reply **to** **Reviewer #1**

Comment:

I have no further acute revision requirements to this manuscript. However, I would still include the voices of some people with dementia on the topic. I find the rationale for non-inclusion to be of limited validity. With regard to potential resource limitations, however, it would be justifiable with reference to the need for further research on the topic, including this target group.

Response:

Thank you very much for your time involved in reviewing the manuscript and your invaluable comments. In line with your comment, we also believe that research related to PwD and relatives should be explored. However, based on the limited resources and considering that professionals are the primary providers of care services in long-term care facilities, we focused on professionals primarily. Subsequent research may further explore the perceptions of PwD and relatives. Thank you again for your comments and for making us think further about the subsequent research!

Reply to Reviewer #2

Dear Ms. Jennifer Bray,

Thank you very much for your time involved in reviewing the manuscript and your invaluable comments to help us improve our manuscript. We appreciate your clear and detailed feedback and hope the explanation has fully addressed your concerns. In this letter, we discussed each of your comments individually along with our corresponding responses. To facilitate this discussion, our responses to your comments are highlighted in blue, and modifications or additions are highlighted in red. You can see a clear copy (without highlighted changes) of the revised manuscript in “Main document” and an edited version of the revised manuscript in “Main document – marked copy”.

Comment 1:

Abstract, lines 30-33 – the sentence ‘Most published...is limited’ is very long and confusing. I can’t work out what it is saying. Which of the following is correct?
Most published meta-synthesis of dementia care focus on hospital or home settings, or long-term care facilities. Most reviews about dementia care only focus on a single outcome...
Or

Most published meta-synthesis of dementia care focus on hospital or home settings. When focusing on long-term care facilities, most reviews about dementia care only focus on a single outcome... It's probably worth rewriting and splitting the sentence to make it clearer.

Response 1: Thank you very much for pointing this out. We have rewritten this sentence. (Line 30-32) "...Most published meta- synthesis of dementia care focus on hospital or home settings. When focusing on long-term care facilities, most reviews about dementia care only focus on a single outcome..."

Comment 2:

Lines 103-106 – the sentence 'Professionals need...quality of life' is also very long and confusing. Do you mean:

Professionals need to take a holistic approach to providing care to the people living with dementia, depending on the situation. Dementia care should include...

Or

Professionals need to take a holistic approach to providing care to the people living with dementia.

Depending on the situation, dementia care should include...

Is the phrase 'depending on the situation' actually needed?

Response 2: Thank you very much for pointing this out. We have revised the sentence to make it clear. (Line 103-104) "...Professionals need to take a holistic approach to providing care to the people living with dementia. Depending on the situation, dementia care should include..."

Comment 3:

Line 106 – I don't 'their' is needed. You can probably just say 'to promote well-being'

Lines 106-110 – I think this sentence might sound better if it is slightly reworded to something like:

'However, knowledge-practice gaps about dementia in long-term care facilities have been revealed by several studies, such as quality of life still needing to be improved,[11] while feeding difficulties,[12] pain management,[13] and other disease-related care needs require further attention.'

Line 110 – I don't think 'the' is needed. You can probably just say 'Besides, behavioral'

Lines 114-115 – it might be better to say 'to provide personalized care and interact with people in long-term' – it avoids referring to people at 'patients' and 'them'

Line 130 – rather than 'a certain number of' I'd suggest saying 'a limited number of' or 'a few'

Line 137 – 'holistic, but' might be better than 'holistic, and'

Lines 193-195 – this sentence might sound better as 'Citation tracing will also be conducted, with the reference lists of included studies being screened to identify studies that were missed in the search process.'

Line 205 – 'a pilot test' might be better than 'the pilot test'

Line 218 – 'will be rated' rather than 'would be rated'

Lines 233-234 – 'data from 5-10 percent of the included' rather than 'data of 5-10 percent included'

Response 3: Thank you very much for your detailed comments. We have reviewed and revised all of the expression issues raised in your comments, and we also have reviewed the entire manuscript.

VERSION 3 – REVIEW

REVIEWER	Bray, Jennifer University of Worcester
REVIEW RETURNED	09-Oct-2023
GENERAL COMMENTS	Thank you for addressing my previous review comments. You will be pleased to know that I have no further comments about your article.